# Linear Pseudospectral Method with Chebyshev Collocation for Optimal Control Problems with Unspecified Terminal Time

Yang Li [ID], Wanchun Chen and Liang Yang *[ID]

School of Astronautics, Beihang University, Beijing 100191, China
* Correspondence: yangliang.buaa@hotmail.com

**Abstract:** In this paper, a linear Chebyshev pseudospectral method (LCPM) is proposed to solve the nonlinear optimal control problems (OCPs) with hard terminal constraints and unspecified final time, which uses Chebyshev collocation scheme and quasi-linearization. First, Taylor expansion around the nonlinear differential equations of the system is used to obtain a set of linear perturbation equations. Second, the first-order necessary conditions for OCPs with these linear equations and unspecified terminal time are derived, which provide the successive correction formulas of control and terminal time. Traditionally, these formulas are linear time varying and cannot be solved in an analytical manner. Third, Lagrange interpolation, whose supporting points are orthogonal Chebyshev–Gauss–Lobatto (CGL), is employed to discretize the resulting problem. Therefore, a series of analytical correction formulas are successfully derived in approximating polynomial space. It should be noted that Chebyshev approximation is close to the best polynomial approximation, and CGL points can be solved in closed form. Finally, LCPM is applied to the air-to-ground missile guidance problem. The simulation results show that it has high computational efficiency and convergence rate. A comparison with the other typical OCP solvers is provided to verify the optimality of the proposed algorithm. In addition, the results of Monte Carlo simulations are presented, which show that the proposed algorithm has strong robustness and stability. Therefore, the proposed method has potential to be onboard application.

**Keywords:** optimal control problems; linear Chebyshev pseudospectral method; unspecified terminal time; missile guidance

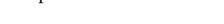



## 1. Introduction

In the middle of 20th century, optimal control began to form and develop as a new discipline. During these decades of development, optimal control has been gradually applied in some engineering fields such as aerospace [1–3], transportation [4], chemical industry [5] and so on. Among them, aerospace is one of the most widely used fields of optimal control. In fact, many problems in aerospace engineering can be regarded as optimal control problems (OCPs) with unspecified final time, which have attracted much attention. The particularity of this kind of problem is that the control and final time are highly coupled, and they need to be solved with high accuracy. For instance, satellite orbit transfer, launch vehicle boost phase guidance and missile terminal guidance can be regarded as OCPs with free final time. Therefore, it is necessary and meaningful to develop an efficient method to address them.

In recent years, many scholars have studied the pseudospectral methods for solving nonlinear OCPs [6–12]. For pseudospectral method, the state and control variables need to be approximated by interpolation polynomials and derivative terms are represented by numerical differentiation. Thus, the dynamic equations can be formulated to a set of algebraic equations on discrete nodes. Therefore, a continuous OCP can be transformed into a nonlinear programming (NLP) problem. Pseudospectral methods are usually separated into Chebyshev, Legendre, Gauss and Radau pseudospectral methods according to discrete

nodes [13–17]. In essence, the pseudospectral method is a direct method. However, highly accurate costate can be approximated. Meanwhile, the first-order necessary conditions (KKT conditions) of NLP that are obtained by the pseudospectral method are equivalent to that of OCPs. Therefore, its optimality has theoretical justification. Unfortunately, the pseudospectral methods require third-party NLP solvers, which occupies a large amount of memory and computational resources. In addition, if the initial value is bad, the convergence rate of the algorithm will be slow. Therefore, pseudospectral methods are usually used for off-line optimization [18–20].

To implement onboard, some scholars aim to improve the computational efficiency of optimal control algorithms and make their code as simple as possible. In [21], an optimization algorithm named model predictive static programming (MPSP) is proposed. This method solves OCPs with terminal constraints and fixed final time. This method has been used in several practical problems [22,23]. Then, Maity et al. extended MPSP to solve the free final time problems [24,25]. In addition, QS-MPSP developed on the basis of MPSP is an improved method [26–28]. In [29], a model predictive control method LGPMPC based on the pseudospectral method is proposed. This method is also used to deal with the OCPs with fixed terminal time. This method combines the idea of quasi-linearization and pseudospectral method, so that the solution procedure no longer depends on the third-party NLP solvers. Additionally, compared with MPSP, LGPMPC has higher computational efficiency. Therefore, this method is suitable for online optimization. So far, LGPMPC has made further development. For instance, an entry guidance method based on LGPMPC is proposed in [30]. In [31], LGPMPC is extended to solve the piecewise continuous OCPs. However, LGPMPC does not consider the variation in terminal time, so it cannot solve the OCPs with unspecified terminal time.

At present, some methods such as the quasi-linearization method and gradient method have the potential to solve the unspecified terminal time OCPs online. The gradient method can be divided into first-order gradient method (FOGM), second-order gradient method (SOGM) and conjugate gradient method (CGM) [32]. It should be noted that both quasi-linearization method and gradient method are iterative algorithms, which approach the optimal solution gradually by iterating the initial value. Some studies have been previously made. In [33], an algorithm based on the improved quasi-linearization technique is proposed to solve the OCPs with unspecified final time. Yang proposed a two-stage gradient method to solve two-stage OCPs with uncertain switching time [34]. A multisystem gradient method was proposed in [35] to solve the integrated OCPs of multiple systems with free terminal time, which is on the foundation of FOGM. In [36], a CGM with pseudospectral collocation scheme is proposed, which is successfully applied to the landing guidance of rockets. Although these methods are successful, we found that if other orthogonal polynomials are involved, the computational efficiency can be further improved and the analytical correction formulas can be further simplified.

In this work, a linear Chebyshev pseudospectral method (LCPM) is proposed for solving OCPs with unspecified terminal time. The main contributions can be summarized as follows. First, by linearizing the dynamic system, the first-order necessary conditions for the discrete form of unspecified terminal time OCPs can be obtained. Second, the correction formulas of control and final time satisfying first-order necessary conditions are derived. The solution to the original problem is transformed into successively solving a set of linear differential equations containing costate variables. Next, the obtained differential equations are discretized on the Chebyshev–Gauss–Lobatto (CGL) points. As a result, a series of analytical correction formulas are successfully derived in approximating polynomial space. The significant differences between this work and previous work are as follows: 1. The variation in terminal time is considered, and the correction formula of terminal time is derived. 2. CGL points are used to discretize the differential equation, and its expansion is very close to the best polynomial approximation under infinite norm. In addition, CGL points can be solved in closed form without using numerical techniques. Finally, the guidance problem of air-to-ground missile is used to test the proposed method. The simulation

results show that LCPM has high computational efficiency and fast convergence rate. In comparison with the results of GPOPS-II, the optimality of LCPM is comprehensively confirmed. In addition, Monte Carlo simulation results show that LCPM can still converge in the presence of random uncertainty, which verifies the robustness and stability of the algorithm. Conclusively, this method has potential to be applied onboard.

The paper is organized as follows. Section 2 shows the first-order necessary conditions of OCPs with unspecified terminal time. In Section 3, the derivations of updating control and final time are presented. The simulation results and discussion are given in Section 4. Finally, the conclusion is given in Section 5.

## 2. Problem Formulation

In this section, the general form and first-order necessary conditions of nonlinear OCPs with unspecified terminal time are given.

Consider a general nonlinear dynamic system whose differential equations can be written as

$$\dot{x} = f(x(t), u(t), t) \tag{1}$$

where $x \in R^n$ is the state vector, $u \in R^m$ is the control vector and $t \in R$ is the time variable. The hard terminal constraints are

$$\psi\left(x(t_f)\right) = 0 \tag{2}$$

Here, $\psi$ is the terminal constraint functions and $t_f$ is the unspecified final time. The performance index is

$$J = \phi\left[x(t_f), t_f\right] + \int_{t_0}^{t_f} L(x, u, t) dt \tag{3}$$

where $t_0$ is the initial time. Equations (1)–(3) describe an OCP with unspecified terminal time. The goal is to find an optimal control and terminal time to minimize the performance index.

Using Taylor expansion and neglecting higher-order terms, the nonlinear differential equations are linearized as

$$\dot{x} = Ax + Bu \tag{4}$$

where $A = \frac{\partial f}{\partial x}$, $B = \frac{\partial f}{\partial u}$.

To solve OCPs iteratively, the state and control variables are expressed as

$$\begin{aligned} x &= x_p - \delta x \\ u &= u_p - \delta u \end{aligned} \tag{5}$$

Here, $x_p$ and $u_p$ represent nominal state and control variables. $\delta x$ and $\delta u$ are the deviations from the state and control. Then, Equation (4) can be rewritten as

$$\dot{x} = (x_p - \delta x)\prime = A(x_p - \delta x) + B(u_p - \delta u) \tag{6}$$

Select $L = \frac{1}{2}u^T R u$, the augmented performance index of the system can be expressed as

$$J_a = \phi\left[x(t_f), t_f\right] + \int_{t_0}^{t_f} \left[\frac{1}{2}(u_p - \delta u)^T R(u_p - \delta u) + \lambda^T(f - \dot{x})\right] dt \tag{7}$$

Hamiltonian function can be defined as

$$H = L + \lambda^T f = \frac{1}{2}(u_p - \delta u)^T R(u_p - \delta u) + \lambda^T(A(x_p - \delta x) + B(u_p - \delta u)) \tag{8}$$

where $\lambda$ is the costate variable.

The modified state and control variables are expected to satisfy the first-order necessary conditions. According to the first-order necessary conditions, it is easy to find that

$$\frac{\partial H}{\partial u} = \frac{\partial H}{\partial(u_p - \delta u)} = R(u_p - \delta u) + B^T\lambda = 0$$
$$\Rightarrow \delta u = u_p + R^{-1}B^T\lambda \tag{9}$$

$$\dot{\lambda} = -\frac{\partial H}{\partial x} = -\frac{\partial H}{\partial(x_p - \delta x)} = -A^T\lambda \tag{10}$$

According to (4), (6) and (9), we have

$$\delta\dot{x} = A\delta x + B\delta u = A\delta x + B\left(u_p + R^{-1}B^T\lambda\right) \tag{11}$$

The transversality condition is

$$\lambda(t_f) = \frac{\partial\phi}{\partial x(t_f)} + \frac{\partial\psi^T}{\partial x(t_f)}v \tag{12}$$

where $v$ is a Lagrange multiplier vector corresponding to terminal constraints. There are two cases of the terminal value of the costate variables. For unconstrained terminal state variable components $x_j(t_f)$, the corresponding costate variables are

$$\lambda_j(t_f) = \frac{\partial\phi}{\partial x_j(t_f)} \tag{13}$$

For constrained terminal state variable components $x_j(t_f)$, the corresponding costate variables are

$$\lambda_j(t_f) = \frac{\partial\phi}{\partial x_j(t_f)} + v_j \tag{14}$$

It can be seen that $\lambda_j(t_f)$ is known when $x_j(t_f)$ is unconstrained, and $\lambda_j(t_f)$ is unknown when $x_j(t_f)$ is constrained.

Since the terminal time is not fixed, the following additional condition needs to be considered.

$$\left(\frac{\partial\phi}{\partial t_f} + H(t_f)\right)_{t_f} = 0 \tag{15}$$

Equations (6), (9), (10), (12) and (15) constitute the first-order necessary conditions for the OCPs with unspecified terminal time.

## 3. Linear Chebyshev Pseudospectral Method

To solve original OCPs with unspecified terminal time, it is necessary to find the optimal control and terminal time that satisfy the first-order necessary conditions. The thought of LCPM is to iteratively make corrections on control and final time through first-order necessary conditions in linear perturbation equations so as to approach the optimal solution gradually.

In this section, the control and terminal time update strategies are derived first. Then, the derivation of LCPM is provided. Finally, the implementation steps of this method are given.

### 3.1. Control and Terminal Time Update Strategies

According to the derivation in Section 2, the correction of the control can be determined by (9). Therefore, it is only necessary to derive the terminal time correction here.

According to (7), the performance index can be written as

$$J_a = \phi\left[x(t_f), t_f\right] + \int_{t_0}^{t_f}\left[L(x,u,t) + \lambda^T(t)f(x,u,t) - \lambda^T\dot{x}\right]dt \tag{16}$$

Considering the differential variations in final time $t_f$, the differential of (16) is

$$
\begin{aligned}
dJ_a &= \frac{\partial\phi}{\partial t_f}dt_f + \frac{\partial\phi}{\partial x(t_f)}dx(t_f) + (L + \lambda^T f - \lambda^T\dot{x})dt_f \\
&+ \int_{t_0}^{t_f}\left[\frac{\partial L}{\partial x}\delta x + \lambda^T\frac{\partial f}{\partial x}\delta x + \frac{\partial L}{\partial u}\delta u + \lambda^T\frac{\partial f}{\partial u}\delta u - \lambda^T\delta\dot{x}\right]dt
\end{aligned}
\tag{17}$$

Integrating (17) by parts, we can get

$$
\begin{aligned}
dJ_a &= \frac{\partial\phi}{\partial t_f}dt_f + \frac{\partial\phi}{\partial x(t_f)}dx(t_f) + (L + \lambda^T f - \lambda^T\dot{x})dt_f - (\lambda^T\delta x)_{t=t_f} \\
&+ (\lambda^T\delta x)_{t=t_0} + \int_{t_0}^{t_f}\left[\left(\frac{\partial L}{\partial x} + \lambda^T\frac{\partial f}{\partial x} + \dot{\lambda}^T\right)\delta x + \left(\frac{\partial L}{\partial u} + \lambda^T\frac{\partial f}{\partial u}\right)\delta u\right]dt
\end{aligned}
\tag{18}$$

Because $x(t_0)$ is given, we have $\delta x(t_0) = 0$. The variation $\delta x$ in $x$ has the significance of keeping time fixed, hence the differentiation and variation in $x$ have the following relations

$$dx(t_f) = \delta x(t_f) + \dot{x}(t_f)dt_f \tag{19}$$

Therefore, $\delta x\left(t_f\right) = dx\left(t_f\right) - \dot{x}\left(t_f\right)dt_f$ can be obtained. Substituting it into (18)

$$
\begin{aligned}
dJ_a &= \frac{\partial\phi}{\partial t_f}dt_f + \frac{\partial\phi}{\partial x(t_f)}dx(t_f) + (L + \lambda^T f - \lambda^T\dot{x})dt_f - \lambda^T dx(t_f) \\
&+ \lambda^T\dot{x}dt_f + \int_{t_0}^{t_f}\left[\left(\frac{\partial L}{\partial x} + \lambda^T\frac{\partial f}{\partial x} + \dot{\lambda}^T\right)\delta x + \left(\frac{\partial L}{\partial u} + \lambda^T\frac{\partial f}{\partial u}\right)\delta u\right]dt
\end{aligned}
\tag{20}$$

If only the differential change $dt_f$ is considered, the change in the performance index due to $dt_f$ is

$$dJ_a|_{\text{due to } dt_f} = \left(\frac{\partial\phi}{\partial t_f} + L + \lambda^T f\right)dt_f \tag{21}$$

From (21), we can choose

$$dt_f = -\varepsilon\left(\frac{\partial\phi}{\partial t_f} + L + \lambda^T f\right) = -\varepsilon\left(\frac{\partial\phi}{\partial t_f} + H(t_f)\right) \tag{22}$$

where $\varepsilon$ is a positive constant. Substituting (22) into (21), we have

$$dJ_a|_{\text{due to } dt_f} = -\varepsilon\left(\frac{\partial\phi}{\partial t_f} + H(t_f)\right)^2 \tag{23}$$

It can be seen that unless $dt_f$ is zero, (23) is negative. Therefore, the $dJ_a$ caused by $dt_f$ will decrease until $\left(\frac{\partial\phi}{\partial t_f} + H\left(t_f\right)\right) = 0$, which also satisfies the first-order necessary condition (15). It can be found that $dt_f$ will reduce the performance index until the optimal solution is reached.

Now the modified forms of control and terminal time have been obtained, as shown in (9) and (22), respectively. However, to figure out $\delta u$ and $dt_f$, the costate variable $\lambda$ needs to be solved first.

Equations (10) and (11) can be written as

$$dJ_a|_{\text{due to } dt_f} = -\varepsilon\left(\frac{\partial\phi}{\partial t_f} + H(t_f)\right)^2 \tag{24}$$

The OCP has been transformed into a two-point boundary value problem. By solving (24), the costate variables $\lambda$ can be solved; thus, the corrections of control and terminal time can be obtained.

### 3.2. Derivation of LCPM

According to the derivation in the previous section, it is known that the key to solving this problem is to solve Equation (24). However, (24) is a set of differential equations, which cannot be solved in an analytical manner. To address this problem and develop a more efficient algorithm, we hope to transfer them into a set of algebraic equations. Thus, Lagrange interpolation polynomials and differential approximation matrices are used to approximately replace variables and differential terms in (24), respectively. The analytical formula can be obtained. The derivation of the algorithm is given below.

In this method, the interpolation nodes used are CGL points, which are in the interval of $[-1, 1]$. Therefore, it is necessary to transform the time domain of the system $\left[t_0, t_f\right]$ to $[-1, 1]$.

$$t = \frac{t_f - t_0}{2}\tau + \frac{t_f + t_0}{2} \tag{25}$$

The performance index is

$$J = \phi\left[x(1), \tau_f\right] + \frac{t_f - t_0}{2}\int_{-1}^{1}\left[\frac{1}{2}\left(u_p - \delta u\right)^T R\left(u_p - \delta u\right)\right]d\tau \tag{26}$$

Equation (11) becomes

$$\delta\dot{x} = \frac{t_f - t_0}{2}A\delta x + \frac{t_f - t_0}{2}B\delta u \tag{27}$$

Differential Equation (24) is transformed to

$$\begin{bmatrix}\delta\dot{x}\\\dot{\lambda}\end{bmatrix} = \begin{bmatrix}\frac{t_f - t_0}{2}A & \frac{t_f - t_0}{2}BR^{-1}B^T\\0 & -\frac{t_f - t_0}{2}A^T\end{bmatrix}\begin{bmatrix}\delta x\\\lambda\end{bmatrix} + \frac{t_f - t_0}{2}\begin{bmatrix}Bu_p\\0\end{bmatrix} \tag{28}$$

The state, control and costate variables can be approximated as

$$\begin{aligned}\delta x^N(\tau) &= \sum_{l=0}^{N}\delta x(\tau_l)\phi_l(\tau)\\\delta u^N(\tau) &= \sum_{l=0}^{N}\delta u(\tau_l)\phi_l(\tau)\\\lambda^N(\tau_k) &= \sum_{l=0}^{N}\lambda(\tau_l)\phi_l(\tau)\end{aligned} \tag{29}$$

The interpolation nodes used here are CGL points, which are the extreme points of N-order Chebyshev polynomials $T_N(\tau) = \cos\left(N\cos^{-1}\tau\right)$. CGL points are defined as

$$\tau_l = \cos(\pi(N - l)/N),\ l = 0, 1, \cdots, N \tag{30}$$

According to the properties of Lagrange interpolation polynomials, we have

$$\phi_l(\tau_k) = \begin{cases} 1, & l = k \\ 0, & l \neq k \end{cases} \tag{31}$$

$$\delta x^N(\tau_k) = \delta x_k;\ \delta u^N(\tau_k) = \delta u_k;\ \lambda^N(\tau_k) = \lambda_k \tag{32}$$

The derivatives of state and costate variables can be represented by the differential approximation matrices as

$$
\delta \dot{x}^N(\tau_k) = \sum_{l=0}^{N} D_{kl} \delta x_l
$$
$$
\dot{\lambda}^N(\tau_l) = \sum_{l=0}^{N} D_{kl} \lambda_l
$$

(33)

where $D_{kl}$ is $N \times (N+1)$ matrix, $k = 1, \cdots, N$, $l = 0, 1, \cdots, N$. $D_{kl}$ can be obtained by taking the derivative of Lagrange polynomials at CGL points.

$$
D_{kl} = \left\{ \sum_{\substack{m=0 \\ m \neq l}}^{N} \left[ \prod_{\substack{j=0 \\ j \neq l, m}}^{N} (\tau_k - \tau_j) \right] \right\} \Bigg/ \left[ \prod_{\substack{j=0 \\ j \neq l}}^{N} (\tau_l - \tau_j) \right]
$$

(34)

Since $x(t_0)$ is given, we have $\delta x(t_0) = 0$. The state and costate variables can be expressed as

$$
\delta x_l = \left( \delta x_1^T \ \cdots \ \delta x_N^T \right)
$$
$$
\lambda_l = \left( \lambda_0^T \lambda_1^T \ \cdots \ \lambda_N^T \right)
$$

(35)

By substituting (33) into (28), a set of linear algebraic equations can be obtained.

$$
\begin{cases}
\sum\limits_{l=0}^{N} D_{kl} \delta x_l - \frac{t_f - t_0}{2} \left( A_k \delta x_k + B_k R^{-1} B_k^T \lambda_k \right) = \frac{t_f - t_0}{2} B_k u_{pk} \\
\sum\limits_{l=0}^{N} D_{kl} \lambda_l + \frac{t_f - t_0}{2} A_k^T \lambda_k = 0
\end{cases}
$$

(36)

where $k = 1, \cdots, N$. (36) can be expressed in matrix form

$$
S^{xx} \delta x + S^{x\lambda} \lambda = M^x
$$
$$
S^{\lambda\lambda} \lambda = M^\lambda
$$

(37)

where

$$
S^{xx} = \begin{bmatrix} D_{11} - \frac{t_f - t_0}{2} A_1 & D_{12} & \cdots & D_{1N} \\ D_{21} & D_{22} - \frac{t_f - t_0}{2} A_2 & \cdots & D_{2N} \\ \vdots & \vdots & \ddots & \vdots \\ D_{N1} & D_{N2} & \cdots & D_{NN} - \frac{t_f - t_0}{2} A_N \end{bmatrix}_{Nn \times Nn}
$$

(38)

$$
S^{x\lambda} = \begin{bmatrix} 0 & -\frac{t_f - t_0}{2} B_1 R^{-1} B_1^T & 0 & \cdots & 0 \\ 0 & 0 & -\frac{t_f - t_0}{2} B_2 R^{-1} B_2^T & \cdots & 0 \\ \vdots & \vdots & \vdots & \ddots & \vdots \\ 0 & 0 & 0 & \cdots & -\frac{t_f - t_0}{2} B_N R^{-1} B_N^T \end{bmatrix}_{Nn \times (N+1)n}
$$

(39)

$$
S^{\lambda\lambda} = \begin{bmatrix} D_{10} & D_{11} + \frac{t_f - t_0}{2} A_1^T & D_{12} & \cdots & D_{1N} \\ D_{20} & D_{21} & D_{22} + \frac{t_f - t_0}{2} A_2^T & \cdots & D_{2N} \\ \vdots & \vdots & \vdots & \ddots & \vdots \\ D_{N0} & D_{N1} & D_{N2} & \cdots & D_{NN} + \frac{t_f - t_0}{2} A_N^T \end{bmatrix}_{Nn \times (N+1)n}
$$

(40)

$$
M^x = \frac{t_f - t_0}{2} \left( B_1 u_{P1} B_2 u_{P2} \ \cdots \ B_N u_{PN} \right)_{Nn}^T
$$
$$
M^\lambda = \left( 0 \ 0 \ \cdots \ 0 \ 0 \right)_{Nn}^T
$$

(41)

The number of equations is $2Nn$. Suppose that the number of state variables constrained by the terminal is $h$, then the $\delta x_N(k), k = 1, \cdots, h$ is known. In addition, $(n - h)$ terminal costate variables $\lambda_N(k); k = h + 1, \cdots, n$ corresponding to the unconstrained states can be solved by (13). Obviously, the number of unknowns is $2Nn$. Therefore, the system has a unique solution.

The costate variables can be solved by (37), and the update of control and terminal time can be obtained from (9) and (22).

$$u = u_p - \delta u = -R^{-1}B^T\lambda \tag{42}$$

$$t_f = t_{fp} + dt_f = t_{fp} - \varepsilon\left(\frac{\partial\phi}{\partial t_f} + L + \lambda^T f\right) \tag{43}$$

*3.3. Implementation Steps of LCPM*

LCPM solves the OCPs with free final time by iteratively updating the control and terminal time. First, the current control and final time are used to integrate the system dynamic equations, then the information of trajectory and terminal error are obtained. Finally, the control and final time are corrected using this information until it is close to the optimal solution. The flow of the algorithm is as follows.

Step 1: Select initial control $u_0$ and final time $t_{f0}$.
Step 2: Use the current control and final time to integrate the dynamic equations and record the information of trajectory and terminal error $d\psi$.
Step 3: Use the information of trajectory and terminal error to solve Equation (37), substitute the obtained costate variables into (42) and (43). Then, the updated control and terminal time can be obtained. To avoid the algorithm divergence caused by a large change in the terminal time $t_f$, we add a measure to restrict $t_f$ in the algorithm. If $\left|dt_f\right| > k \cdot t_f$, we choose $\left|dt_f\right| = k \cdot t_f$, while the sign stays the same. Here, $k$ is a positive constant. In this paper, we select $k = 2.5\%$.
Step 4: Take the updated control and final time as the current control and final time, return to step 2, and judge whether the values of the terminal error and Equation (15) meet the requirements. If so, the algorithm ends; if not, it continues.

The flow chart of the algorithm is shown in Figure 1. $\varepsilon_\psi$ and $\varepsilon_{t_f}$ are the convergence criteria of the algorithm, and the specific values are given in Equation (47).

Through the above steps, the OCPs with unspecified terminal time can be solved. It should be noted that the initial control, initial terminal time, parameters $\varepsilon$, $k$ and convergence criteria for terminal error need to be provided before the algorithm starts.

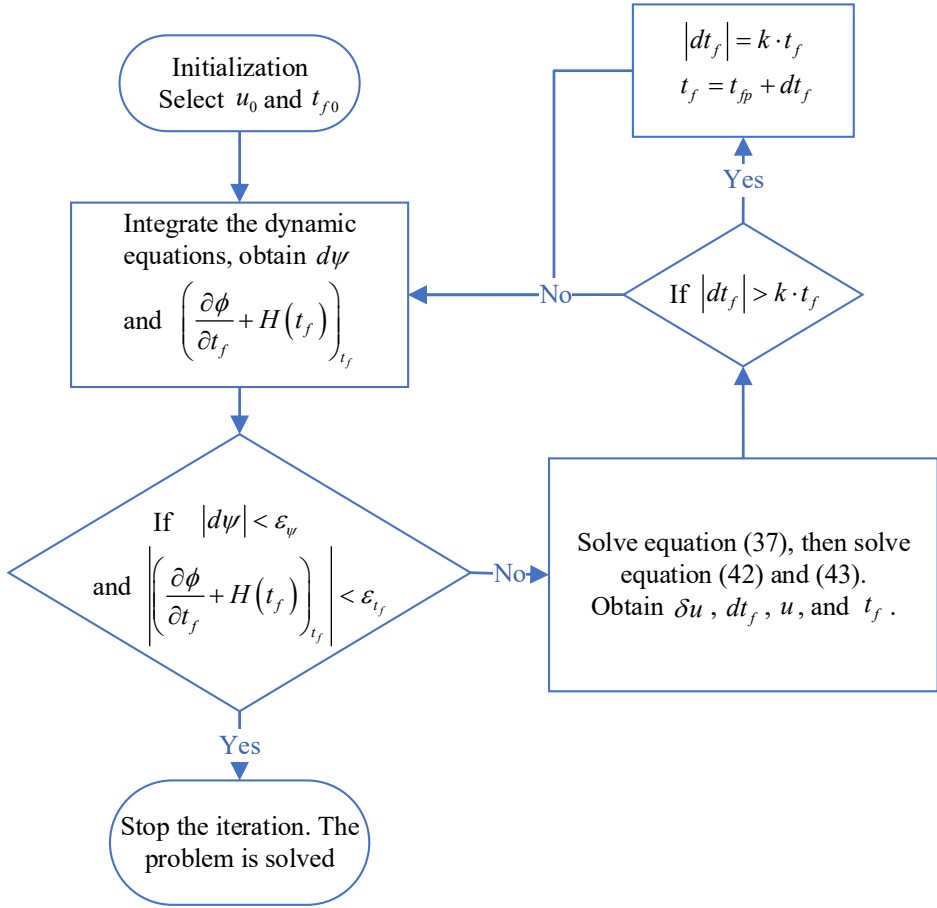

**Figure 1.** Flow chart of the algorithm.

## 4. Numerical Examples

In this section, the guidance problem of air-to-ground missiles, which can be regarded as an OCP with unspecified terminal time, is used to test LCPM. The simulation example used is similar to that in reference [22], but the difference is that the terminal time in [22] is fixed, while the terminal time in our work is free. To verify the adaptability of the method in different situations, three different conditions are selected for simulation. The three simulations have the same initial conditions but different terminal constraints. Next, the simulation results of LCPM are compared with that of GPOPS-II and FOGM. GPOPS-II is a commonly used pseudospectral optimization software, which adopts Radau pseudospectral technology. As described in the introduction, the traditional pseudospectral methods are often used in off-line optimization. Therefore, GPOPS-II is used here to provide the optimal solution to verify the optimality of LCPM. FOGM is an efficient optimization algorithm [32], which is used to verify the efficiency of LCPM. Additionally, the LCPM in this paper is compared with some of our previous work [35–37]. Finally, the convergence process of the algorithm is given to discuss the convergence rate of the algorithm. All simulations are performed on a personal computer with Core i5-1135G7 (2.4 GHz) processor using 2018b MATLAB.

### 4.1. Dynamic Model and Simulation Parameters

The system dynamic equations are

$$
\begin{aligned}
\dot{v} &= \frac{-D}{m} - g \sin \gamma & \dot{\gamma} &= \frac{-a_z - g \cos \gamma}{v} \\
\dot{\psi} &= \frac{a_y}{v \cos \gamma} & \dot{x} &= v \cos \gamma \cos \psi \\
\dot{y} &= v \cos \gamma \sin \psi & \dot{z} &= v \sin \gamma
\end{aligned}
\tag{44}
$$

where $x$, $y$, $z$ are three coordinates of the missile in space, $v$ is the speed, $\gamma$ is flight path angle, $\psi$ is heading angle and $g$ is the acceleration of gravity. $D$ is the drag, which can be referred to in [38]. $a_z$ and $a_y$ are command accelerations of the missile, that is, the control variables.

The order of magnitude difference among different variables may lead to the instability of numerical calculation. Therefore, it is necessary to normalize the variables.

$$
\begin{array}{cccc}
v_n = \frac{v}{v*} & \gamma_n = \frac{\gamma}{\gamma*} & \psi_n = \frac{\psi}{\psi*} & x_n = \frac{x}{x*} \\
y_n = \frac{y}{y*} & z_n = \frac{z}{z*} & a_{zn} = \frac{a_z}{a_z*} & a_{yn} = \frac{a_y}{a_y*}
\end{array}
\tag{45}
$$

The subscript "$n$" means the normalized variables, while the superscript "*" denotes the normalized constants. In addition, the performance index is selected as

$$
J = \frac{1}{2}\int_{t_0}^{t_f} u^T u\, dt
\tag{46}
$$

The initial control and initial final time used in the simulation are obtained by proportional navigation, which can be referred to in [22]. The relevant parameters used in simulations are provided in Tables 1–3 show the initial conditions and terminal constraints in different cases. It should be noted that the correction of $t_f$ depends on the parameter $\varepsilon$. Hence, an appropriate $\varepsilon$ makes the algorithm converge faster. We recommend the value of $\varepsilon$ to be 0.02–0.1.

**Table 1.** Simulation parameters.

| Parameters | Value |
|---|---|
| Normalizing velocity | 600 m/s |
| Normalizing angle, $(\gamma*, \psi*)$ | (50 deg, 50 deg) |
| Normalizing coordinates, $(x*, y*, z*)$ | (5 km, 5 km, 5 km) |
| Normalizing acceleration $(a_z*, a_y*)$ | $g = 9.81$ m/s$^2$ |
| Mass of missile | 150 kg |
| Surface area, $s_m$ | 0.0324 m$^2$ |
| $\varepsilon$ in case 1, 2, 3 | (0.05, 0.078, 0.078) |

**Table 2.** Initial conditions.

| | $v(m/s)$ | $\gamma(\circ)$ | $\psi(\circ)$ | $(x,y,z)$ $(km)$ |
|---|---|---|---|---|
| Case 1 | 510 | 15 | 15 | (0, 2, 6) |
| Case 2 | 510 | 15 | 15 | (0, 2, 6) |
| Case 3 | 510 | 15 | 15 | (0, 2, 6) |

**Table 3.** Terminal constraints.

| | $v(m/s)$ | $\gamma(\circ)$ | $\psi(\circ)$ | $(x,y,z)(km)$ |
|---|---|---|---|---|
| Case 1 | \ | −65 | −40 | (10, 1, 0) |
| Case 2 | \ | −70 | −35 | (11, 1.5, 0) |
| Case 3 | \ | −75 | −40 | (10, 2, 0) |

The "\" in Table 3 means that no constraint is imposed on the terminal velocity. The terminal time is unspecified, which can be automatically calculated by the algorithm.

### 4.2. Simulation Results and Discussion

First, the simulation results of LCPM are compared with that of FOGM and GPOPS-II. In the simulation, the fourth-order Runge–Kutta method is used to integrate the trajectory,

and the number of integration steps is 100. The number of CGL points selected is 10. When the algorithm ends, the terminal errors need to meet the following requirements

$$
\begin{aligned}
&\left|\delta x_f\right| < 1(\mathrm{m}); \ \left|\delta y_f\right| < 1(\mathrm{m}); \ \left|\delta z_f\right| < 1(\mathrm{m}) \\
&\left|\delta \gamma_f\right| < 0.001(\mathrm{rad}); \ \left|\delta \psi_f\right| < 0.001(\mathrm{rad}) \\
&\left|\left(\frac{\partial \phi}{\partial t_f} + H(t_f)\right)_{t_f}\right| < 0.1
\end{aligned}
\tag{47}
$$

Figure 2 shows the trajectories of the missile in three cases. It can be seen that the missiles successfully attacked the designated terminal points. Figures 3 and 4 are the flight path angle and heading angle histories, respectively. Obviously, all constraints of terminal state variables are enforced for three methods. Figures 5 and 6 show the command accelerations, which are the control variables in the solving process. They are smooth and continuous. In the three cases, the state and control curves obtained by LCPM coincide with the optimal solution generated by GPOPS-II, which verifies the optimality of LCPM. In addition, the stability of the algorithm is verified by applying it to different situations.

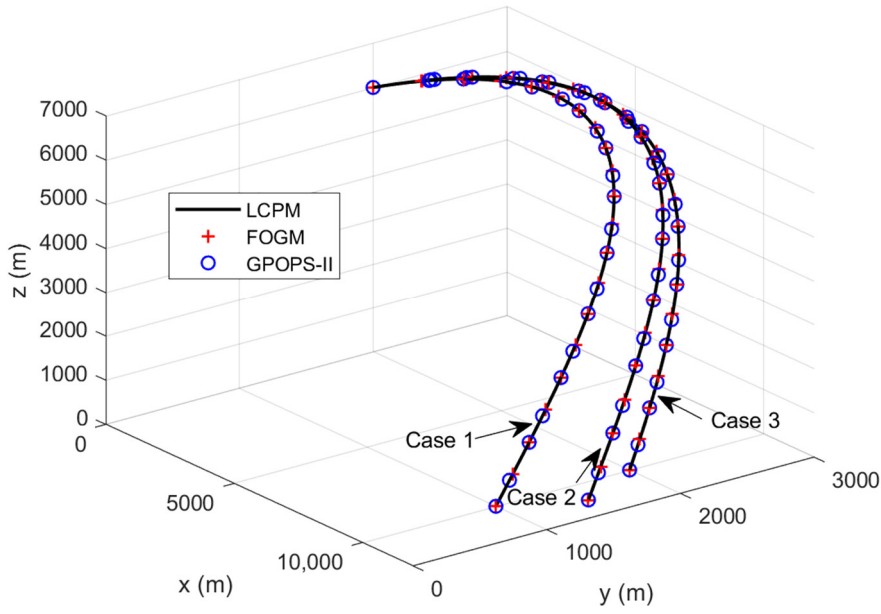

**Figure 2.** Comparison of missile 3D trajectories.

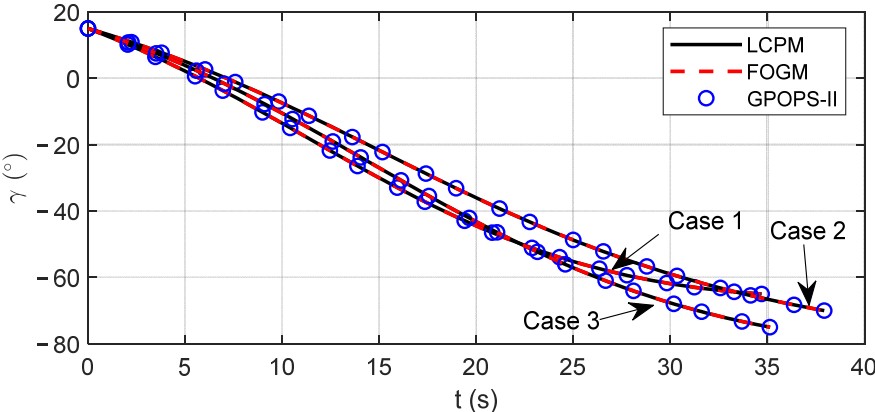

**Figure 3.** Comparison of flight path angle.

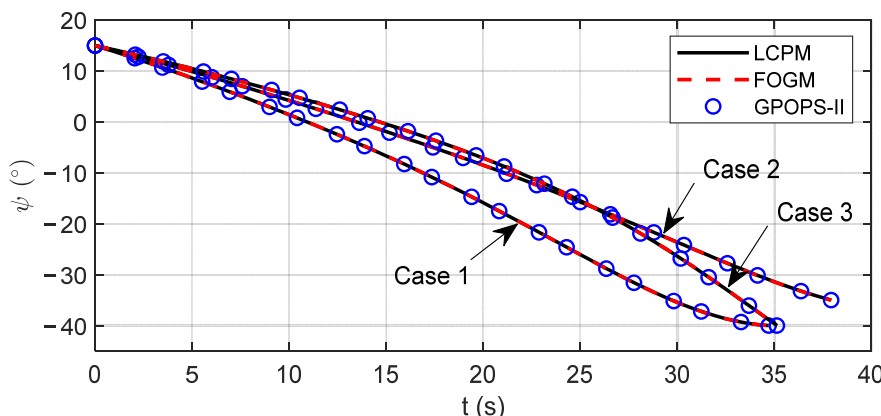

**Figure 4.** Comparison of heading angle.

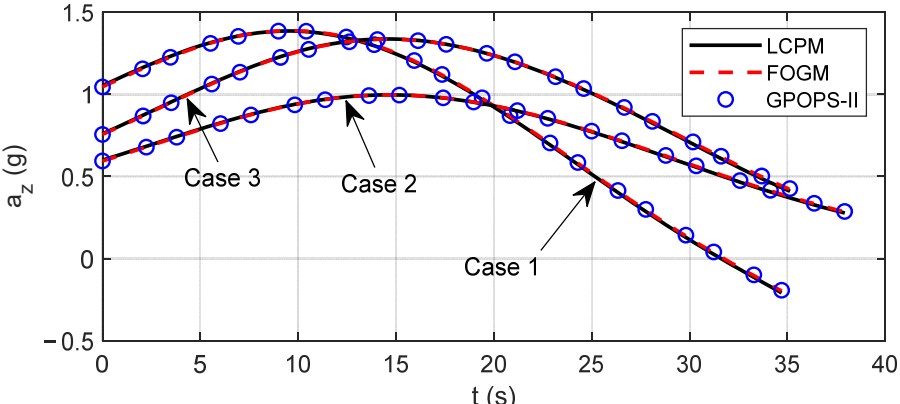

**Figure 5.** Comparison of control $a_z$.

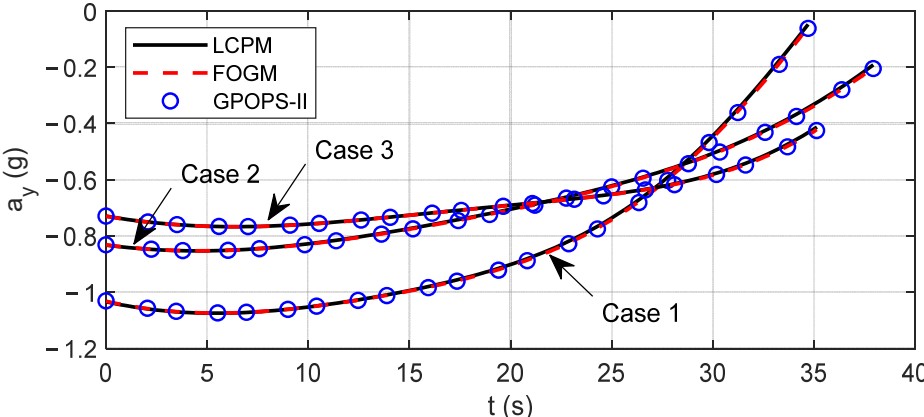

**Figure 6.** Comparison of control $a_y$.

As mentioned in the introduction, FOGM is an efficient algorithm for solving the OCPs with unspecified terminal time. Therefore, time consumption for LCPM and FOGM is provided to verify the efficiency of LGPM. Actually, the time consumption of these two algorithms is mainly composed of two parts, the one is trajectory integration (step 2), and the other is control update (step 3). The trajectory integral process of the two algorithms is similar, so the time consumption is similar. As the core part of the algorithms, the control update time consumption of the two methods is quite different. From Table 4, it can be found that the trajectory integral time consumption of LCPM and FOGM is similar, while the control update time consumption of LCPM is significantly less than FOGM. The reason

is that LCPM removes the calculating process of many special integrals which is necessary for FOGM to update the control. It is easy to conclude that the proposed algorithm has high computational efficiency. Compared with FOGM, LCPM can improve the calculation efficiency by about 20–30%. If more specialized computer and programming languages are used, the time consumption will be further reduced.

**Table 4.** Time consumption of LCPM and FOGM.

| Average Time Consumption of a Single Iteration | Trajectory Integration (s) | | Control Update (s) | | Efficiency Improvement of LCPM |
|---|---|---|---|---|---|
| | LCPM | FOGM | LCPM | FOGM | |
| Case 1 | 0.0196 | 0.0188 | 0.0018 | 0.0078 | 19.6% |
| Case 2 | 0.0191 | 0.0198 | 0.0019 | 0.0083 | 25.3% |
| Case 3 | 0.0187 | 0.0201 | 0.0017 | 0.0086 | 28.9% |

Next, LCPM is compared with our previous work [35–37]. In [35], a multisystem gradient method (MSGM) is proposed. To apply MSGM to the simulation example of this paper, we consider its number of systems as one. In [36], a pseudospectral collocation conjugate gradient method is proposed to solve the OCPs with hard terminal constraints and fixed final time. To compare PCCG with LCPM, the terminal time obtained by LCPM is selected as the terminal time of PCCG. In [37], a successive Chebyshev pseudospectral convex optimization (SCPCO) method is proposed. For the sake of fairness, these three methods are simulated in the same computing environment as LCPM. Because the integral prediction time consumption of several methods is basically the same, the time consumption of the control update is compared. The comparison results are shown in Table 5.

**Table 5.** Time consumption of LCPM and other three methods.

| Average Time Consumption of a Control Update | LCPM | MSGM | PCCG | SCPCO |
|---|---|---|---|---|
| Case 1 | 0.0018 | 0.0031 | 0.0076 | 0.48 |
| Case 2 | 0.0019 | 0.0030 | 0.0077 | 0.49 |
| Case 3 | 0.0017 | 0.0028 | 0.0079 | 0.49 |

As can be seen from Table 5, LCPM has higher computational efficiency than the other three methods on this problem. However, the other three methods have their own advantages. For example, MSGM can be used to solve the optimization problems of multiple systems, and PCCG can be used to solve the problems with general performance indexes. As for SCPCO, because it needs to use a third-party solver, the computational efficiency is relatively low. Nevertheless, SCPCO can solve problems with control and state constraints, and has good global optimality. Through this comparison, it shows that LCPM has advantages in solving the problems given in Section 2.

The control update speed of LCPM is closely related to the number of CGL points. With the increase in CGL points, computational efficiency will decrease. The number of CGL points should be carefully selected by users. Table 6 shows the control update time consumption of LCPM with a different number of CGL points. In general, the number of CGL points can be selected as 5–15.

**Table 6.** Time consumption of LCPM with different CGL points.

| Control Update Time | Number of CGL Points | | | | |
|---|---|---|---|---|---|
| | 6 | 8 | 10 | 15 | 20 |
| Case 1 | 0.0014 | 0.0015 | 0.0018 | 0.0029 | 0.0046 |
| Case 2 | 0.0013 | 0.0016 | 0.0019 | 0.0031 | 0.0047 |
| Case 3 | 0.0013 | 0.0015 | 0.0017 | 0.0030 | 0.0048 |

In order to reveal the convergence process of the algorithm as clear as possible, the variation in the terminal error and terminal time with the number of iterations is plotted.

Because terminal error is a vector, the infinite norm of the vector is selected to measure terminal error.

$$\text{terminal error} = \|d\psi(t_f)\|_{\infty} \tag{48}$$

Figures 7 and 8 show the variation in the terminal error and terminal time, respectively. It can be seen that in the previous iterations, the convergence trend of terminal error is not obvious, while the terminal time quickly approaches the optimal solution. In the later iterations, the terminal error converges rapidly, but the terminal time changes slightly. That is consistent with the fact that great changes in the terminal time lead to slow convergence for the terminal error. When the terminal time is near the optimal one, the terminal error has fast convergence rate. It can be seen from Figures 7 and 8 that the algorithm can converge after only a few iterations. Therefore, LCPM has a fast convergence rate. After the last iteration, the terminal error and Equation (15) both meet the requirements of (47). The normalized terminal errors in three cases are given in Table 7.

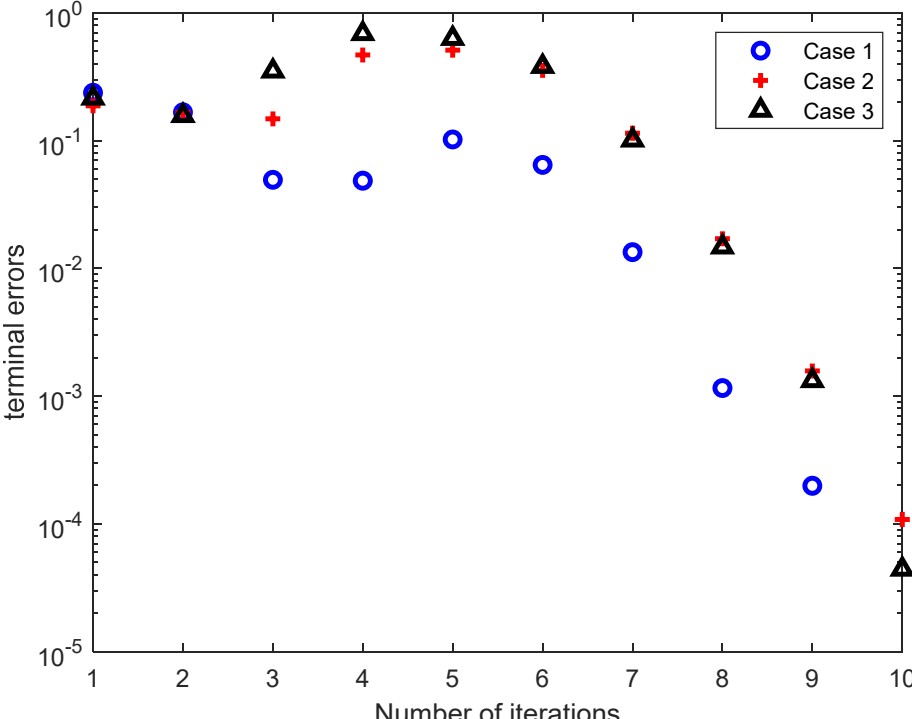

**Figure 7.** Variation in terminal error in three cases.

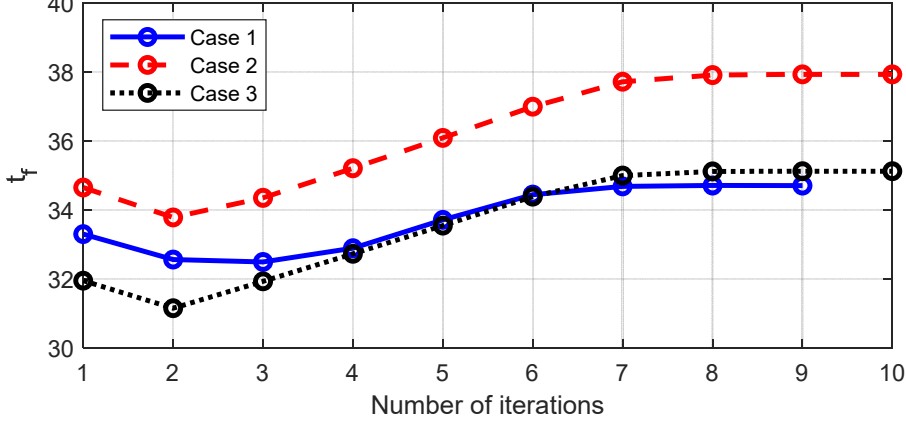

**Figure 8.** Variation in terminal time in three cases.

**Table 7.** Terminal errors in three cases.

|         | $d\gamma_f$ | $d\psi_f$ | $dx_f$ | $dy_f$ | $dz_f$ |
|---------|------------|-----------|--------|--------|--------|
| Case 1 | $3.927 \times 10^{-5}$ | $-3.024 \times 10^{-5}$ | $4.105 \times 10^{-5}$ | $1.085 \times 10^{-4}$ | $1.986 \times 10^{-4}$ |
| Case 2 | $2.030 \times 10^{-5}$ | $-7.520 \times 10^{-5}$ | $2.545 \times 10^{-5}$ | $7.365 \times 10^{-5}$ | $1.083 \times 10^{-4}$ |
| Case 3 | $4.408 \times 10^{-5}$ | $-4.389 \times 10^{-5}$ | $2.778 \times 10^{-5}$ | $7.602 \times 10^{-6}$ | $1.097 \times 10^{-6}$ |

In the actual flight process, the vehicle suffers from the uncertainty of atmospheric density and aerodynamic coefficient, which are the most critical factors for the success of flight mission. This requires the algorithm to have good stability and robustness. To verify the robustness of the proposed method, Monte Carlo simulation is carried out, in which the simulation conditions are the same as that in case 1. The uncertainty of the aerodynamic coefficients and atmospheric density follows the $3\sigma$ principle of normal distribution. The simulation parameters are given in Table 8. In total, 500 Monte Carlo simulations results are presented in Figures 9 and 10, Table 9.

**Table 8.** Dispersion of parameters.

| Parameters | $3\sigma$ Range | Distribution |
|------------|-----------------|--------------|
| $\delta\rho$ | $\pm15\%$ | Gaussian |
| $\delta C_L$ | $\pm15\%$ | Gaussian |
| $\delta C_D$ | $\pm15\%$ | Gaussian |

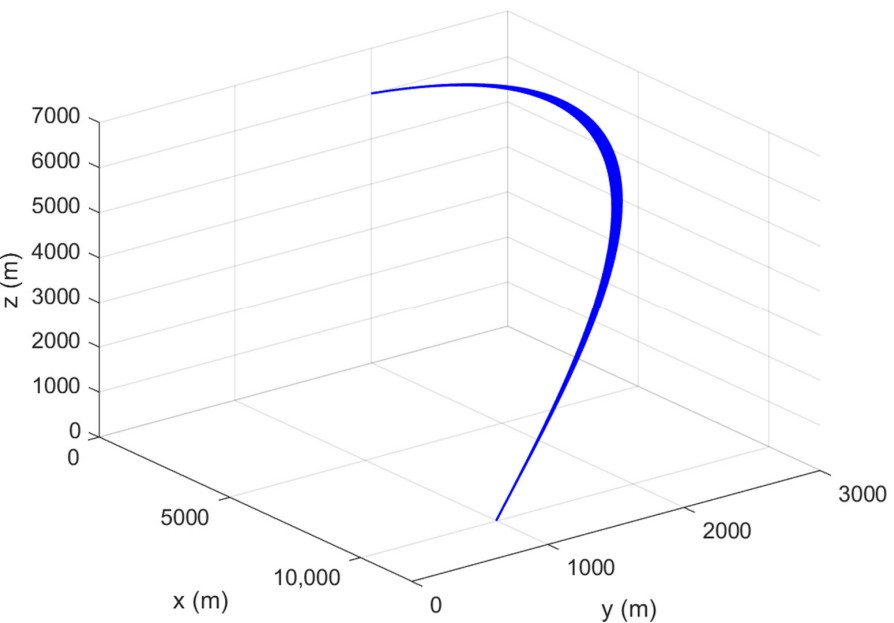

**Figure 9.** Monte Carlo simulation results of missile trajectory.

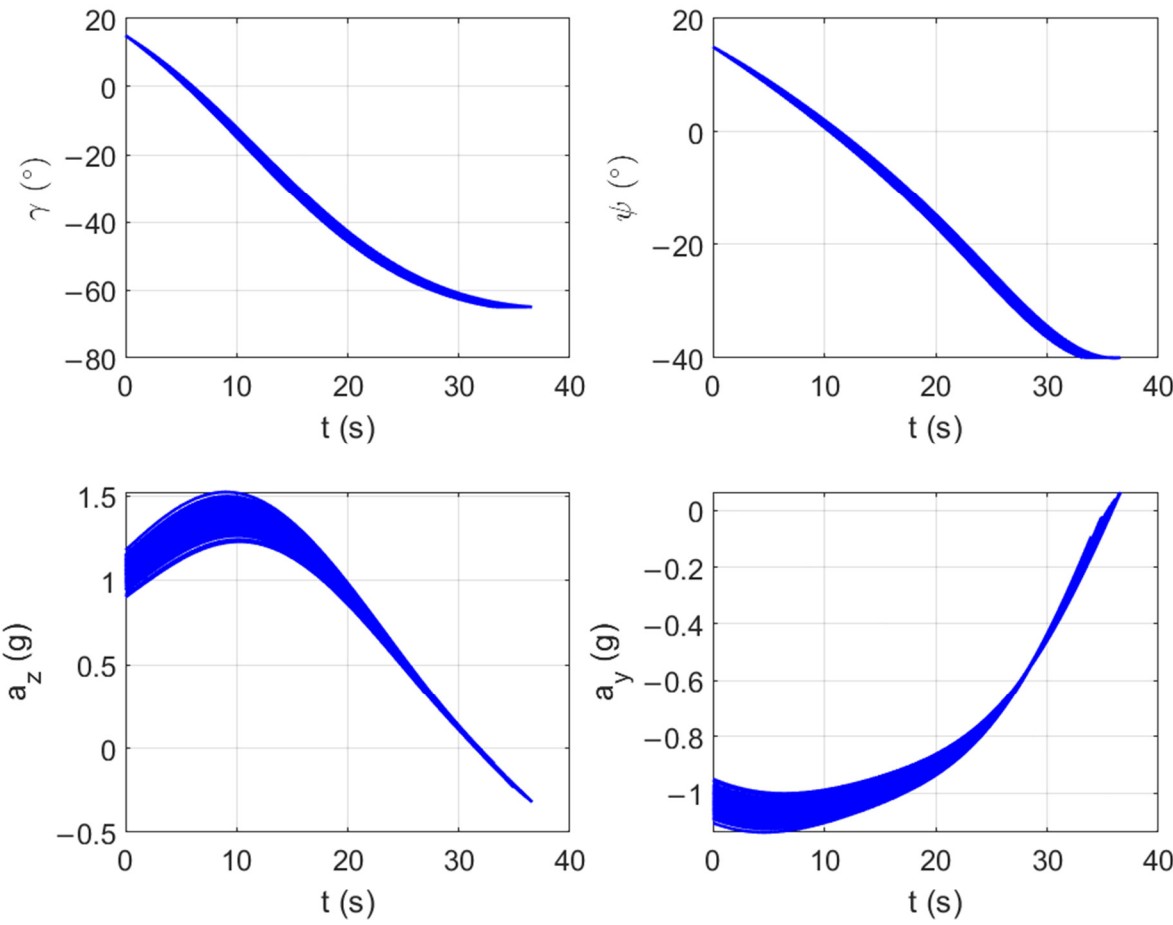

**Figure 10.** Monte Carlo simulation results of state and control variables.

**Table 9.** Mean and variance of terminal errors.

| Terminal Errors | Mean | Variance |
|---|---|---|
| $d\gamma_f$ | $5.5441 \times 10^{-6}$ | $6.8281 \times 10^{-10}$ |
| $d\psi_f$ | $6.4341 \times 10^{-5}$ | $3.4546 \times 10^{-9}$ |
| $dx_f$ | $8.8958 \times 10^{-5}$ | $5.0744 \times 10^{-9}$ |
| $dy_f$ | $8.9497 \times 10^{-6}$ | $2.3554 \times 10^{-9}$ |
| $dz_f$ | $-4.4885 \times 10^{-5}$ | $2.3276 \times 10^{-8}$ |

The missile trajectory histories of Monte Carlo simulations are shown in Figure 9, and the state and control variables are shown in Figure 10. It is obvious that the proposed method can adapt to all kinds of random situations. From the mean and variance given in Table 9, it can be inferred that the terminal errors are closely distributed around zero. Therefore, the algorithm has strong robustness even when large uncertainty and dispersion are involved.

## 5. Conclusions

A linear Chebyshev pseudospectral method is proposed for solving the nonlinear OCPs with free final time. The control and final time correction formulas are derived theoretically. The solution to the original problem is transformed into successively solving a set of linear equations. Then, Chebyshev polynomials are used to discretize them so as to successfully drive analytical correction formulas for control and terminal time. As we all know, real-time applications require high efficiency and convergence rate of the algorithm. To verify the performance of the method, it is applied to the guidance problem of

air-to-ground missiles. It can be seen from the simulation results that the proposed method not only has optimality, but also has high computational efficiency and a fast convergence rate. Additionally, the stability and robustness of the proposed method are verified by Monte Carlo simulations. Therefore, we believe that this method has the potential to be applied online in practical engineering problems.

**Author Contributions:** Conceptualization, Y.L. and L.Y.; methodology, Y.L.; software, Y.L.; validation, Y.L.; formal analysis, Y.L. and W.C.; investigation, Y.L. and L.Y.; resources, W.C.; data curation, L.Y.; writing—original draft preparation, Y.L.; writing—review and editing, L.Y.; visualization, Y.L.; supervision, W.C.; project administration, W.C.; funding acquisition, L.Y. All authors have read and agreed to the published version of the manuscript.

**Funding:** This research was supported by the National Natural Science Foundation of China under Grant 62003019 and by the Young Talents Support Program funded by Beihang University under Grant YWF-21-BJ-J-1180.

**Institutional Review Board Statement:** Not applicable.

**Informed Consent Statement:** Not applicable.

**Data Availability Statement:** Not applicable.

**Conflicts of Interest:** The authors declare no conflict of interest.

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
