# Peer review of "Linear Pseudospectral Method with Chebyshev Collocation for Optimal Control Problems with Unspecified Terminal Time"

_aerospace, doi:10.3390/aerospace9080458_

Round 1
Reviewer 1 Report
The authors have presented the problem nicely. They provided detailed mathematical derivation and simulation results to support their claims. I want to mention a few points as follows.
1. In Introduction the authors mentioned about MPSP. But they forgot to mention the improved variety of it, which is designed to reduce the computational burden. This variety is 'QS-MPSP'. I request the authors they must include the following references to make their literature review complete and meaningful.
''Angle-Constrained Terminal Guidance using Quasi-Spectral Model Predictive Static Programming''
''State and Input Constrained Missile Guidance using Spectral Model Predictive Static Programming''
2. line 55: To implement
3. How did you get Eq 22 from 21. dJa is missing. Explain the importance of \eta.
4. In Table 3, the terminal velocity column must have something meaningful.
5. Please provide the improvement achieved by LCPM over the other method (for ex execution time) in a table with percentage.
Reviewer 2 Report
This paper's technical content appears to be fine, but there is a significant lack of literature review and comparison. Try to carefully address the following comments in a major revision and make the modification explicitly visible in the revised manuscript.
The literature review must be improved. The statement regarding the MPSP only addressing fixed final time is wrong. In recent years, Prof. Arnab Maity cited in [23] published multiple papers about MPSP with free final time. Please complete the investigation.
The simulation study is not comprehensive. Comparisons with other methods including your own works, e.g., [32], should be made, showing differences, especially in computational efficiency.
Make both detailed theoretical analysis and numerical comparison with the following paper, where you have to draw solid conclusions to justify the publication of this paper.
Li, Yang, Wanchun Chen, and Liang Yang. "Successive Chebyshev pseudospectral convex optimization method for nonlinear optimal control problems." International Journal of Robust and Nonlinear Control 32.1 (2022): 326-343.
Round 2
Reviewer 2 Report
The responses given by the authors are rather disappointing. The requested comparisons can be done, but the authors did not attempt to do them and the reasons provided by the authors for not doing them are not really justified.
For fixed final time only approaches, a final time can be specified for comparison purposes to enable the implementation. Moreover, you could have compared it with [35].
Regarding the authors' previous work that solves a similar class of problems, they do not even bother citing it. The requested analyses and comparisons are still unfortunately missing. The revisions should be visible in the revised manuscript.
One more chance is suggested by the reviewer if the authors still wish to address the concerns.
Author Response
We are very sorry for our response in last revision and thank you very much for giving us another opportunity to revise our manuscript. We have made a revision according to the reviewer’s comments. Please see the attachment.

Round 3
Reviewer 2 Report
The authors added the comparison as requested. Now the paper can be accepted.